# Subepidermal Low-Echogenic Band—Its Utility in Clinical Practice: A Systematic Review

**DOI:** 10.3390/diagnostics13050970

**Published:** 2023-03-03

**Authors:** Alin Codrut Nicolescu, Sinziana Ionescu, Ioan Ancuta, Valentin-Tudor Popa, Mihai Lupu, Cristina Soare, Elena-Codruta Cozma, Vlad-Mihai Voiculescu

**Affiliations:** 1“Agrippa Ionescu” Emergency Clinical Hospital, 011773 Bucharest, Romania; 2Surgery Department, “Carol Davila” University of Medicine and Pharmacy, 050474 Bucharest, Romania; 3Surgery Department, “Prof. Dr. Al. Trestioreanu” Oncology Institute Bucharest, 022328 Bucharest, Romania; 4Rheumatology Department, “Dr. I. Cantacuzino” Clinical Hospital, 020475 Bucharest, Romania; 5Dermatology Department, Center for Morphologic Study of the Skin MORPHODERM, “Victor Babes” University of Medicine and Pharmacy, 300041 Timisoara, Romania; 6Department of Dermatology, MEDAS Medical Center, 030447 Bucharest, Romania; 7Dermatology Department, Elias University Emergency Hospital, 011461 Bucharest, Romania; 8Pathophysiology Department, University of Medicine and Pharmacy of Craiova, 200349 Craiova, Romania

**Keywords:** high-frequency ultrasonography, sub-epidermal low-echogenic band, echogenicity, non-invasive diagnosis, atopic dermatitis

## Abstract

High-frequency ultrasonography (HF-USG) is a relatively new imaging method that allows the evaluation in a non-invasive manner of the skin layers and skin appendages. It is a diagnostic tool with increasing usefulness in numerous dermatological pathologies. High reproducibility, non-invasiveness and short diagnostic time make this method an increasingly used tool in dermatological practice. The subepidermal low-echogenic band is a relatively newly described parameter that seems to be a marker not only of intrinsic and extrinsic skin aging, but also of inflammatory processes taking place at the skin level. This systematic review aims to evaluate the role that SLEB has in the diagnosis and monitoring of the treatment of some inflammatory and non-inflammatory dermatological conditions, as well as its utility as a disease marker.

## 1. Introduction

High-frequency ultrasonography (HF-USG) is a non-invasive imaging technique used since the second half of the 20th century (1979) to evaluate skin layers and structures. This is accomplished by using devices with high frequencies (over 15Mhz) which allow the examiner to increase the resolution of ultrasonographic images, sacrificing at the same time the depth of penetration [1]. Although this is a method known for over a century, only in the last decade scientific attention has been directed towards it and the possibility of its use in numerous inflammatory or tumoral dermatological diseases [2]. The possibility of assessing the thickness of epidermis and dermis, their echogenicity and density, as well as the visualization of a new parameter, namely, the subepidermal low-echogenic band (SLEB), opened the possibility of using this method not only for the descriptive purpose and evaluation of tumor margins, but also as the interface of the processes of inflammation and aging (intrinsic or photoinduced) that occur at the skin level [3]. SLEB is described as a hypoechoic, well-defined band, located immediately under the epidermis layer. The two layers of the dermis (papillary and reticular), although different in composition and organization of the collagen fibers (in the case of the papillary dermis being more loosely packed), are very difficult to differentiate ultrasonographically. However, SLEB, by its hypoechoic appearance, seems to correspond to lax papillary dermis [4]. Variations in its thickness and echogenicity are closely related to photo-aging processes, through the degradation of collagen fibers at this level, but also to the invasion of this area by inflammatory cells and the accumulation of water molecules (edema). Thus, SLEB has become an increasingly frequent parameter in cutaneous HF-USG studies, regardless of the studied disease [1,3]. This work represents the first systematic review that highlights the role and usefulness of SLEB in the diagnosis, staging, monitoring of disease activity, as well as in determining the response to topical therapies in various dermatological diseases.

## 2. Materials and Methods

This systematic review is based on the protocol elaborated by Moher et al. regarding ”The Preferred Reporting Items for Systematic reviews and Meta-Analyses (PRISMA)” [5].

The investigators performed an advanced search in several international databases (PubMed/Medline, Web of Science, SCOPUS) for articles published during 2012–2022, using the word combination” subepidermal low echogenic band”.

Several inclusion criteria had to be met for articles to be included in this systematic review, namely: 1. the papers were original articles, research letters, or case series that evaluated the role of SLEB in the diagnosis, prognosis, description of the stage of a disease or its role in monitoring different invasive dermatological procedures in physiological processes (such as skin aging) or pathological processes (inflammatory, autoimmune and neoplastic diseases); 2. the selected papers were presented in a language spoken by the authors (English, French or Romanian); 3. the selected papers were published after 1 January 2012; 4. the full text of the articles/case series/research letters could be obtained.

Exclusion criteria included the following: 1. the papers represented reviews, single case reports, letters to the editors, or abstracts presented at scientific meetings; 2. the papers were written in a language not known by any of the investigators; 3. the studies were performed on cell cultures or animals; 4. the studies were published before 1 January 2012; 5. the investigators could not obtain the full-text version of the articles; 6. the papers did not offer sufficient information regarding the role of SLEB in the diagnosis, prognosis, description of diseases or in the monitoring of invasive dermatological procedures.

The selection process of the articles to be included consisted in a preliminary scan of the titles of the articles resulting from the search based on the combination of words mentioned above; then, the next step involved the reading of the abstracts and the full-text articles of the works resulting from the first scan. A database in Microsoft Excel was also created which included the following key information on the selected studies: first author name, year of publication, country where the study was conducted, study design, number of patients included, dermatological conditions studied by ultrasound, frequency of the ultrasound probe used, reporting of the level at which the SLEB measurement was made, pathological changes of the SLEB. In addition, the investigators ensured the scientific quality of the selected studies and assessed the risk of bias by using two essential tools, namely, the Methodological Index for Non-Randomized Observational Studies (MINORS), and the Mixed Methods Appraisal Tool (MMAT) [6,7].

## 3. Results

Figure 1 represents the diagram of the selection process of the 24 articles included in this systematic review. From the initial group of 39 articles resulting from the use of the search key, duplicates and, subsequently, articles whose titles and/or abstracts did not match our research question were eliminated, resulting in 34 articles that were read in full text. Of these, 24 papers met the inclusion criteria and are discussed below (Table 1).

### 3.1. SLEB and Atopic Dermatitis (AD)

A total of three studies evaluated the role of SLEB in establishing the severity of the examined disease and in monitoring therapy efficacy as well as the best moment for switching from a reactive to a proactive treatment in patients with atopic dermatitis (AD) [8,9,10] (Figure 2A,B).

In an observational study, Polanska et al. observed a group of 39 patients (21 women, mean age 26.3 ± 12.8 years old) with AD who underwent reactive (one application/day, between 1 and 6 weeks) and proactive treatment (2–3 applications/weeks) with tacrolimus 0.1%, followed by the evaluation of the group clinically, evaporimetrically and by HF-USG. The patients were evaluated at the initial presentation and then monthly, up to 6 months of treatment, examining both the affected area (right antecubital fold) and an area of apparently normal skin, 10 cm away from the affected area. Regarding the HF-USG measurements, significant differences were found in the size of SLEB, both in the affected skin and in apparently intact skin. Thus, SLEB decreased from 0.227 mm to 0.03 mm after 24 weeks in the affected skin (*p* < 0.001) and from 0.03 mm to 0.00 mm (*p* < 0.001) after 24 weeks, respectively, in the apparently healthy skin. In addition, Polanska et al. observed a decrease in the number of patients in which SLEB was visualized: from 100% to 18% (*p* < 0.001) in the affected skin and from 20.5% to 5.1% (*p* < 0.001) in the healthy skin after 24 weeks of treatment [8].

Another study conducted by Sorokina et al. on a group of 22 children with AD (14 females, mean age 3.7 ± 2.4 years) and a control group of 18 healthy children (11 females, mean age 3.9 ± 2.2 years) evaluated several parameters of skin ultrasonography, both in the lesional skin as well as in the normal skin of the control group and in the healthy perilesional skin of children with AD. The areas analyzed were the cheek, the ulnar region and the popliteal fossa. Regarding the evaluation of SLEB, the presence of SLEB was found in 22% of healthy children, with sizes between 53 microns and 92 microns. In the group of children with AD, SLEB was identified in 100% of the examined lesions and in 77% of the apparently healthy perilesional skin, with thicknesses between 68 microns and 106 microns. The lowest values found in both groups were at the ulnar level [9].

Sabau et al. evaluated, in a group of 10 patients with AD (8 women, mean age 26 years), ultrasound aspects of lesional and nonlesional skin (SLEB, skin thickness and skin intensity), obtaining an average SLEB on the lesional skin of 164 microns and on healthy skin of 13 microns; the average thickness was 1409 mm and 0.8755 mm, respectively, and the average skin intensity was 37.95 and 60.4, respectively. Moreover, they observed that the Disease Life Quality Index (DLQI) correlated with skin thickness (*p* = 0.039, c = 0.657) and skin intensity (*p* = 0.032, c = 0.675), but not with SLEB or disease severity assessed by the SCORAD score (Scoring Atopic Dermatitis) [10].

### 3.2. SLEB and Skin Aging

A number of seven studies evaluated by HF-USG different skin aspects (including SLEB) related to skin aging caused both by the cumulative effect of ultraviolet (UV) radiation and by aging [11,12,13,14,15,16,17].

Crisan et al. evaluated in a study conducted on a group of 160 patients (80 women, mean age = 40.4 ± 21.1), divided into four age categories (under 20 years, 21–40 years, 41–60 years, over 80 years), the ultrasonographic aspects of the skin in three photoexposed areas (dorsal forearm, medial arm, zygomatic arc) and one non-photoexposed area (medial arm). They visualized SLEB in all patients over 20 years old, especially in photoexposed areas, compared to the population under 20 years of age. They also found a significant increase in the thickness of the dermis (*p* = 0.035) in patients in the age range of 21–40 years [11].

Rayner et al. evaluated in a pilot study the test–retest reliability of measuring SLEB, skin thickness and skin intensity in a group of 31 patients (22 women, mean age 83.3 ± 4.23) by means of the intraclass correlation coefficient (ICC) and the Lin’s concordance correlation coefficient (CCC). Rayner et al. assessed the degree of skin aging and the risk of skin tearing. The ultrasonographic characteristics were measured at the level of the mid dorsal forearm and the upper quarter of the lateral lower leg, bilaterally (places characterized by skin aging and tear) and at the half distance between the umbilicus and the left iliac crest (control area). They observed an almost perfect test–retest reliability for SLEB (0.95–0.99) and for skin thickness (0.95–0.99) [12].

The above-mentioned study was continued a few years later by the same group that evaluated 173 patients (123 women, mean age = 87.8 ± 6.7 years) in order to establish the risk of developing purpura at the forearm level, through two separated measurements, at a distance of 6 months. The researchers found that SLEB and skin thickness were characteristics of aged skin that were statistically significantly associated with purpura in multivariable analysis (*p* = 0.014), but not in univariate one (*p* = 0.92). They also observed a positive correlation between SLEB and forearm skin thickness (*p* < 0.01, r = 0.512) [13].

Tedeschi et al. evaluated the SLEB changes after performing two hyaluronic acid injection protocols at the level of the dorsal face of the hand, bilaterally. The study was carried out on 22 women (mean age 50.5 years) who were administered 1 mL of hyaluronic acid weekly for one month and then monthly, for 4 months the first group, and for 9 months the second group. SLEB was measured before the injection and one week after it at the level of the second metacarpal web, and an increase in echogenicity of SLEB was observed in 24% of the participants (*p* < 0.01) after 4 weeks, and in 18% of them after 4 and 10 months (*p* < 0.05) [14].

Wakade et al. evaluated, in a group of 40 women, the impact on SLEB of two aesthetic procedures performed weekly for 4 weeks on the two halves of the face (radiofrequency and chemical peeling with glycolic acid). SLEB was measured bilaterally, both at the level of the external canthus of the eye and at the level of the nasolabial fold. They observed an insignificant decrease in SLEB at the eye level with both types of procedures and a significant decrease at the nasolabial fold level (a decrease in SLEB from 0.32 mm to 0.25 mm on the half of the face subjected to radiofrequency and a decrease from 0.3 to 0.2 mm on the half of the face subjected to chemical peeling) [15].

Mondon et al. evaluated in an ex vivo and in vivo study several histopathological, confocal microscopy and skin ultrasonography aspects of skin aging. They assessed the effects of palmitoyl oligo and tetrapeptides applied on the face and unilateral forearm, compared to those of a placebo. The peptides were applied daily for 2 months to a group of 28 women over 50 years old (mean age = 59 ± 5.4 years). A significant improvement was observed in SLEB thickness (which decreased by 11–14.4% at the forearm level, *p* < 0.01) and SLEB density (which increased by 15% at the forearm level, *p* < 0.01) [16].

Arisi et al. investigated, in a group of eight patients with field cancerization, skin ultrasonography aspects related to the presence of actinic keratosis and photoaged perilesional skin before and after 3 months of cold atmospheric plasma therapy. They highlighted an increase in the density of the dermis and perilesional SLEB (*p* = 0.02) and a decrease in the thickness of the perilesional SLEB (from 223 to 146.5 microns, *p* = 0.04) after the therapy. No statistically significant changes were observed regarding the thickness of the epidermis or dermis before and after the treatment [17].

### 3.3. SLEB and Skin Lymphomas

A number of five studies evaluated SLEB changes and its role in evaluating disease progression and response to treatment in patients with cutaneous lymphomas (mycosis fungoides, folliculotropic mycosis fungoides and Sesary syndrome) [18,19,20,21,22].

Polanska et al. evaluated, in 2017, in a group of 18 patients (5 women, mean age = 52.2 years) with mycosis fungoides stages I-IIA, the changes in SLEB and skin thickness measured by HF-USG before and after treatment with UVA1 and PUVA (7–10 weeks). They observed the presence of SLEB in all the examined lesions, but not in normal skin, regardless of the stage of the disease. Moreover, after the treatment, the disappearance of SLEB was observed in 66% of the cases, as well as a significant decrease in its thickness, from 0.256 mm to 0.064 mm (*p* < 0.001). In addition, after the treatment, a change in echogenicity of the skin was observed, both in the lesions (where it increased from 7.49% to 7.94%) and at the level of apparently intact skin (from 7.8% to 8.46%) (*p* < 0.01) [18].

Later, the same team evaluated, in a group of three patients with mycosis fungoides stage IB (mean age = 48.3 years), the evolution of SLEB after treatment during a period of follow-up of approximately 5 years. Thus, before the initiation of the treatment, a mean SLEB = 0.44 mm was observed, which subsequently decreased to 0.13 mm (*p* = 0.01). A correlation of SLEB with clinical response was also observed, with SLEB disappearing completely in patients with complete response, unlike in those with partial response (SLEB = 0.16mm) [19].

Another study by Polanska et al. in a group of 10 patients (1 woman) with mycosis fungoides highlighted the correlations between the presence of SLEB and that of a histological lymphocytic infiltrate. They observed an average SLEB thickness of 0.488 mm, with higher values in the plaque stage (0.544 mm) compared to the patch stage (0.265mm). Statistically significant positive correlations between the size of SLEB and the size of the lymphocytic infiltrate observed at the histopathological examination (r = 0.994, *p* < 0.01), with higher values of the former, were also detected [20].

Niu et al. evaluated ultrasonographic skin differences (epidermal morphology and thickness, level of inflammatory infiltrate, SLEB thickness, margins and echogenicity) in patients with early mycosis fungoides (19 patients) and psoriasis vulgaris/eczema (48 patients). They observed a lower thickness of the epidermis (*p* < 0.01) and SLEB (*p* = 0.006) in patients with early mycosis fungoides compared to those with psoriasis/eczema. Moreover, the cut-off values for these patients were 0.2375 mm for the thickness of the epidermis (sensitivity 88.9% and specificity 73%) and 0.2655 mm for the thickness of SLEB (sensitivity 55.6% and specificity 90.9%) [21].

The latest study led by Yukun Wang et al. evaluated the margins and echogenicity of SLEB in 26 patients with cutaneous lymphomas (23 with classic mycosis fungoides, 2 with folliculotropic mycosis fungoides and 1 with Sezary syndrome). Both in the patch and in the plaque stages, the presence of SLEB with clearly defined edges and homogeneous echogenicity was observed, ultrasound differences being observed at the level of the epidermal layer, with a homogeneous epidermis in the patch stage and a wavy epidermis in the plaque stage. At the same time, in patients with both folliculotropic and Sezary Syndrome forms, the same aspect of SLEB described above was found, but associated with the presence of hypoechoic perifollicular areas [22].

### 3.4. SLEB and Psoriasis

We identified two studies that evaluated the features of SLEB in patients with psoriasis [23,24] (Figure 2A,C).

Oweczarczyk-Saczonek et al. used HF-USG to monitor the response to treatment with dichiroinositol 1% and 0.25% compared to a placebo in 40 patients with mild psoriasis. They found no statistically significant differences regarding the thickness of the epidermis or SLEB in the two groups compared to the placebo group. However, at the end of the treatment, SLEB was identified in only 25% of the patients, whereas it was visualized in 70% of them at the initiation of the therapy [23]. Regarding the ultrasonographic aspects of psoriasis, Odrzywolek et al. evaluated skin thickness, skin density and SLEB at the level of lesions in 71 patients, observing significantly lower skin densities in patients with psoriasis (*p* = 0.0003), as well as a thickened epidermis (*p* = 0.257). These two parameters were proportionally inversely correlated (*p* = 0.001). Higher SLEB thickness values were also observed in these patients, with the greatest thickness found at the knee level (0.389 mm) [24].

### 3.5. Other Applications of HF-USG in Skin Diseases

Several studies evaluated the usefulness of HF-USG in general and of SLEB in particular in assessing different skin structures in special areas (genital) or in the presence of certain dermatological pathologies (leg ulcer, diabetic ulcer, lichen planus, facial granuloma, psoriasis vulgaris, talar calluses, limb oedema) [25,26,27,28,29,30,31].

Migda et al. evaluated, in a study carried out in 50 women (age between 20–80 years), the ultrasonographic characteristics of the skin and adjacent structures at the level of the mons pubis and the labia majora and minora. They found the presence of SLEB in 80% of the examined patients, all cases identified having previously undergone cosmetic procedures which might have influenced the results [25].

Krause et al. observed ultrasonographic changes in a group of eight patients (four women, mean age = 70.2 ± 11.6 years) with shin ulcer without signs of healing in the last 2 months, for which they decided to use bio-stimulation by laser therapy (two sessions/week). They observed an increase in the granulation tissue measured by HF-USG (from 2.66 ± 1.44 mm to 2.97 ± 0.7mm) and a decrease in SLEB measured in the perilesional tissue (from 0.78 ± 0.3 mm to 0.57 ± 0.37 mm) [26]. Regarding foot ulcers, this time in patients with diabetes, Chao et al. evaluated a group of 19 patients with diabetic ulcers, 35 patients with diabetes but without neuropathy and 33 healthy patients. They observed an increase in the thickness of the plantar epidermis in both groups with diabetes compared to the healthy population (*p* < 0.05). In addition, in both groups SLEB was higher than in the healthy population (*p* < 0.001), with a more pronounced increase in patients with ulcers (64.7%) compared to those without ulcers (11.8%). Moreover, they observed a negative correlation between SLEB thickness and the thickness of the epidermis at the level of the hallux (*p* = 0.002, r = −0.366) [27].

Yazdanparast et al. evaluated changes in healthy, perilesional and lesional skin in 21 patients with lichen planus (13 women, mean age = 47.62 ± 15.36 years), observing the presence of SLEB in 76.2% of the lichen lesions. Moreover, a statistically significant increase in SLEB was observed at the level of the lesions (from 22.64 ± 38.54 microns to 346.33 ± 281.55 microns, *p* < 0.01) compared to healthy skin, but not at the level of the perilesional skin [28].

Another inflammatory lesion evaluated ultrasonographical is the facial granuloma, whose HF-USG aspects were examined by Morgado-Carrasco et al. in five patients. They highlighted the presence of the lesion as a hypoechoic, heterogeneous, poorly defined mass that invaded the dermis and hypodermis, most frequently associated with the presence of SLEB in all the patients [29].

Suehiro et al. also evaluated SLEB thickness and several parameters related to subcutaneous echogenicity in 30 patients (mean age 67 years) with unilateral lymphedema in the context of breast cancer. They found statistically significantly higher values of all measured parameters (skin thickness, SLEB, subcutaneous tissue thickness, subcutaneous echogenicity, subcutaneous echo-free space) (*p* < 0.05) in all measured areas (medial upper arm, lateral upper arm, medial forearm, lateral forearm, dorsum of the hand) compared to the contralateral arm without lymphedema [31].

Talar callosities represent another pathology for which HF-USG is useful. Luna-Bastante et al. evaluated ultrasonographic aspects in a group of four children with talar callosities that can cause diagnostic difficulties. They observed the disappearance of SLEB associated with dermo–hypodermic thickening, allowing a differential diagnosis with respect to other inflammatory lesions associated with a thickening of SLEB [30].

**Table 1 diagnostics-13-00970-t001:** Changes in SLEB thickness and echogenicity in different pathologies.

Year	Study Type	No. of Patients	Studied Disease	Ultrasound Frequency	Is the Level of SLEB Measuring Reported?	Is SLEB Present in Non Lesional Skin?	SLEB Changes	Reference
2015	Interventional study	39	Atopic dermatitis	15–25 MHZ	Yes	Yes	DT	[8]
2019	Observational study	10	Atopic dermatitis	15–25 MHZ	No	Yes	IT, DE	[10]
2020	Observational study	18	Atopic dermatitis	>25 MHZ	No	Yes	IT	[9]
2012	Case series	160	Skin aging	15–25 MHZ	Yes	Yes	IT	[11]
2015	Interventional study	28	Skin aging	>25 MHZ	Yes	Not applied	IT, IE	[16]
2015	Interventional study	22	Skin aging (fillers with hyaluronic acid)	15–25 MHZ	Yes	Not applied	IE	[14]
2016	Interventional study	40	Skin aging (chemical peels and radiofrequency)	>25 MHZ	Yes	Not applied	DT	[15]
2017	Case series	31	Skin aging	15–25 MHZ	No	Yes	IT	[12]
2019	Observational study	173	Skin aging	15–25 MHZ	No	Yes	IT, DE	[13]
2021	Interventional study	12	Aktinic keratosis and photoaging	>25 MHZ	No	Not applied	IT, DE	[17]
2017	Interventional study	18	Cutaneous lymphoma (Mycosis fungoides)	15–25 MHZ	No	No	DT, IE	[18]
2018	Observational study	3	Cutaneous Lymphoma (Mycosis fungoides)	15–25 MHZ	No	No	IT, DE	[19]
2019	Observational study	10	Cutaneous lymphoma	15–25 MHZ	No	No	IT, DE	[20]
2020	Observational study	26	Cutaneous Lymphoma (Mycosis fungoides)	>25 MHZ	No	No	IT, DE	[22]
2021	Case series	67	Cutaneous lymphoma (Mycosis fungoides), psoriasis/eczema	>25 MHZ	No	Not applied	IT, DE	[21]
2012	Observational study	87	Diabetic foot	>25 MHZ	Yes	Yes	IT	[27]
2016	Observational study	30	Arm lymphedema	1–14 MHZ	Yes	Not applied	IT	[31]
2019	Interventional study	50	Vaginal rejuvenation	15–25 MHZ	Yes	No	IT, DE	[25]
2019	Observational study	21	Lichen planus	>25 MHZ	No	Yes	IT	[28]
2020	Interventional study	9	Venous leg ulcer (cold atmospheric plasma treatment)	15–25 MHZ	Yes	Yes	IT	[26]
2021	Observational study	5	Granuloma faciale	15–25 MHZ	No	No	IT, DE	[29]
2021	Case series	4	Talar callosity	1–14 MHZ	Yes	Yes	DT, IE	[30]
2021	Interventional study	46	Psoriasis (D-chiro-inositol treatment)	15–25 MHZ	No	Yes	IT, DE	[23]
2022	Observational study	71	Psoriasis	>25 MHZ	Yes	Yes	IT, DE	[24]

IT, increased thickness (measured in mm, perpendicular to skin surface), DT, decreased thickness (measured in mm, perpendicular to skin surface), IE, increased echogenicity, DE, decreased echogenicity.

## 4. Discussion

HF-USG represents a relatively new, non-invasive imaging method that allows the in vivo evaluation of physiological and pathological aspects of the skin, as well as the influence of endogenous and exogenous factors [11]. This non-invasive imaging technique finds its usefulness more and more often in the evaluation of inflammatory skin diseases, being possible in the future perhaps even to use it as a marker of treatment efficiency. This is supported by studies that show the presence of SLEB in AD in all cases before the start of treatment, followed by a significant decrease of it after approximately 6 months of treatment (*p* < 0.001) [8]. Moreover, in the same dermatological condition, the presence and thickness of SLEB can orient us with a fairly high accuracy in order to establish the optimal moment in which to switch from a reactive treatment to a proactive treatment, in order to reduce relapses, SLEB values being influenced by edema and inflammatory infiltrate at the skin level [8]. At the same time, the thickness of SLEB correlates with the degree of parakeratosis, hyperkeratosis, spongiosis and inflammatory infiltrate, but not with the DLQI and the SCORAD score that assesses the severity of the disease perceived by patient and doctor [10]. Another inflammatory condition associated with ultrasonographic changes is lichen planus, which is associated with an increase in the thickness of the dermis and a decrease in its density on HF-USG, but also with an increase in SLEB thickness, most likely due to edema and inflammatory infiltrate. Thus, HF-USG can be used in patients with this pathology to increase the diagnostic accuracy, but also to monitor the response to treatment [28]. To increase the accuracy of a non-invasive diagnosis, we can also use this method in the diagnosis of facial granulomas, HF-USG allowing their differentiation from other facial lesions, such as cutaneous lupus erythematosus or cutaneous lymphomas [29]. However, in this sense, the characteristic hypoechoic–heterogeneous appearance, with poorly defined edges, is more useful than the presence of SLEB (which probably corresponds to the inflammatory infiltrate present histopathologically), this being a non-specific parameter also found in the inflammatory diseases described above and especially as a result of photoaging to which the skin of the face is subjected [29].

Moreover, versatility, reproducibility, relatively high accuracy, as well as minimal discomfort make HF-USG an effective technique in the evaluation of lesions in the pediatric population, where the percentage of atopic dermatitis is increasing. Thus, although SLEB, is not pathognomonic, it allows the correlation in pediatric patients with edema and inflammatory infiltrate [9]. Moreover, through the obtained images we can appreciate in real time aspects related to skin morphology and pathophysiological mechanisms, which allow us to observe the content of collagen, the orientation and distribution of collagen fibers at the skin level. Thus, the echogenicity of the dermis becomes one of the most important parameters that allow us to examine the ultrasonographic echogenicity, which is influenced by the orientation of the collagen fibers and the water content at this level [10]. Regarding aspects related to skin aging, HF-USG allows the assessment of skin aging characteristics [12]. Thus, SLEB can be perceived as a skin-aging marker, correlating with the thickness of the skin at the forearm level (*p* < 0.01, r = 0.512), an increased SLEB indicating the destruction of collagen fibers at the level of the papillary dermis and their replacing with elastin deposits and glycosaminoglycans. However, no correlation of SLEB with intrinsic aging was observed, suggesting a greater impact in the skin aging process of UV radiation than of aging itself [13]. Thus, it was observed that the echogenicity of SLEB is inversely proportional to skin aging [14]. Moreover, the same parameter, SLEB, can be used in the elderly population to assess the risk of developing a purpura at the level of the forearms, SLEB thickness measured at this level being positively correlated with this risk [13]. The thickness and density of SLEB can also be used to assess the response to treatment of skin with signs of photoaging and field cancerization. Arisi et al. observed an improvement in this parameter after therapy (increase in density and decrease in thickness), as well as an increase in the density of the dermis, both, probably, through collagen synthesis and through the activation of immune system cells [17].

Regarding the usefulness of SLEB in aesthetic medicine, it can be used as an objective marker to assess the collagen remodeling process that occurs after minimally invasive aesthetic procedures, allowing a non-invasive, long-term follow-up of the results [15]. Moreover, Mondon et al. highlighted the fact that HF-USG can be used complementary to confocal reflectance microscopy to assess non-invasively aspects related to the improvement of skin quality (changes in SLEB’s increased echogenicity, decreased thickness and papillary dermis’ increased echogenicity) after topical application of anti-aging dermatocosmetics [16]. Thus, HF-USG can become a useful tool to appreciate the real effectiveness of these products and highlight how they exert their anti-aging effects.

HF-USG is also useful in the management of cutaneous lymphomas, especially type T (mycosis fungoides). SLEB is an objective parameter that allows the appreciation of the degree of lymphocytic inflammatory infiltrate that correlates statistically significantly with histopathological aspects (*p* < 0.01, r = 0.99) and with the stage of the disease [18,22]. This imaging method is also useful in increasing the diagnostic accuracy, compared to main differential diagnoses (psoriasis, non-specific eczema), some studies showing statistically significantly lower values in patients with early mycosis fungoides compared to those with psoriasis/eczema in the thickness of both epidermis and SLEB, which reflects the pathophysiological processes underlying these conditions [21]. However, HF-USG is not a diagnostic method, as the ultrasonographic aspects are not specific to this pathology, and histopathological and immunohistochemical confirmation is still necessary. Ultrasonography, by visualizing SLEB and its modifications along with the evolution of the disease, therefore allows the administration of topical therapy and the assessment of the partial or complete response to treatment, as well as of the residual disease, being thus an indicator for the continuation or interruption of therapy [18,19]. Cutaneous ultrasonography can also be used in the assessment of the skin and adnexal structures at the genital level in the female population, the ultrasound data correlating with the histopathological ones, as HF-USG is able to identify the vulvo–vaginal and cervical anatomical structures, the thickness of the skin and its layers and the accessory glands. Moreover, the identification of SLEB at this level allows the identification of inflammatory pathologies, such as dermatitis, eczema, inverted psoriasis or irritations caused by various local procedures [25].

In the case of leg ulcers, HF-USG can be used to assess the granulation tissue developed before and after the use of topical or interventional therapies so to assess the vascularity of the ulcer base or the perilesional tissue, as well as to visualize de novo epithelization. SLEB measured in these patients, in addition to the phenomena of skin aging and degradation of collagen fibers, may also reflect water retention at the level of the papillary dermis. Thus, a subsequent decrease in SLEB along with the improvement of the appearance of the lesion can be explained by a decrease in edema at this level [26]. This theory is also be supported by the study conducted by Suehiro et al., which highlighted a thicker SLEB in all examined patients with lymphedema [31]. The same theory is also supported by Chao et al., who explained the thicker SLEB in patients with diabetic ulcers and diabetes without neuropathy by the presence of edema in the papillary dermis (clinically manifested also by edema of the lower limbs in these patients) which consequently determined atrophy of the underlying skin, more pronounced in patients with ulcers than in those without ulcers. Thus, the negative correlation between the thickness of the SLEB and that of the epidermis entails numerous pathophysiologic implications for these patients and may in the future be a parameter for assessing the patient risk of developing skin tears [27].

## 5. Conclusions

Important progress has been made in the last decade regarding the histopathological correspondence of SLEB, as well as the translation of its presence into pathophysiological processes. However, SLEB is not a parameter specific for a certain dermatological pathology, being rather associated with collagen degradation processes, inflammation and edema at the level of the papillary dermis. Although it can currently be used to follow the activity of a disease, guide the duration of treatment or assess the response to treatment in the presence of numerous inflammatory diseases, we are still far from relying on HF-USG as the only diagnostic tool for dermatological diseases, the existing studies up to the current time, although with statistically significant results, not having analyzed large, representative groups of patients. Thus, the need for randomized controlled studies in large groups of patients, the presence of clear histopathological correlations, as well as the increase in resolution in order to differentiate the skin layers more accurately are still essential elements necessary for the large-scale applicability of this parameter in clinical practice.

## Figures and Tables

**Figure 1 diagnostics-13-00970-f001:**
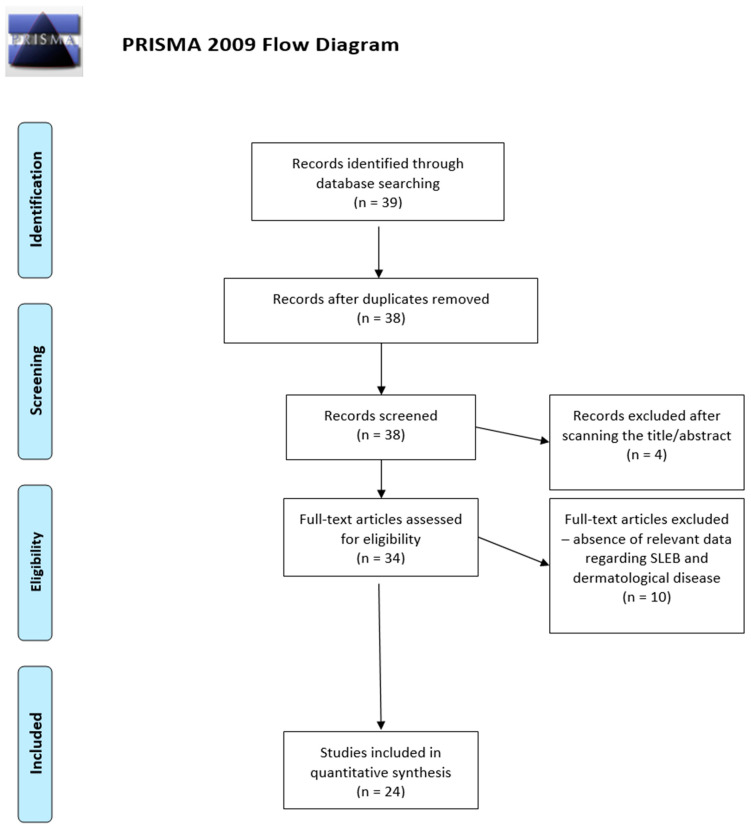
PRISMA 2009 flow diagram depicting the selection process of the articles included in the systematic review [5].

**Figure 2 diagnostics-13-00970-f002:**
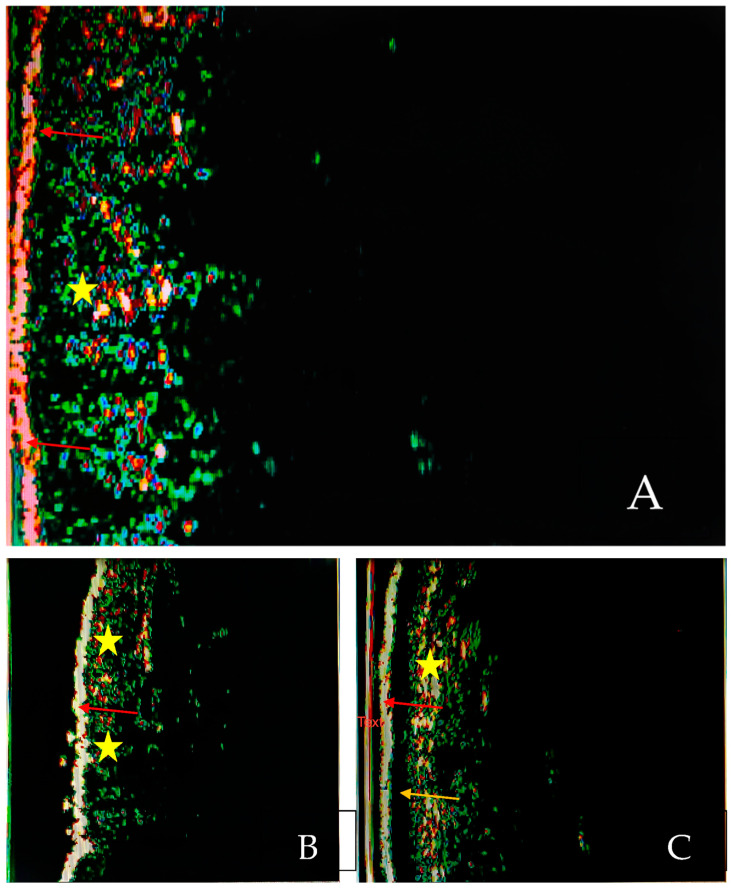
HF-USG aspects of normal skin (**A**) and skin with atopic dermatitis (**B**) and psoriasis (**C**). Visualization of epidermis (red arrows), SLEB (yellow arrow) and dermis (yellow star). The images were captured using a device with a 20 Mhz transducer (personal library of Dr. Alin Nicolescu).

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
