# Peer review of "Subepidermal Low-Echogenic Band—Its Utility in Clinical Practice: A Systematic Review"

_diagnostics, 2023, doi:10.3390/diagnostics13050970_

Round 1

Reviewer 1 Report

Authors reported a systematic review to evaluate the role of subepidermal low echogenic band (SLEB) not only as a marker of intrinsic and extrinsic skin aging, but also in the diagnosis and monitoring of the treatment of some inflammatory and non-inflammatory dermatological conditions.

STATUS: ACCETTABLE FOR PUBBLICATION PENDING MINOR REVISIONS

General considerations:

The article is a systematic review. The work is interesting, and the paper is very well-written. Despite extensive research, I did not find any systematic reviews on this topic in the literature. Therefore, I recommend its publication, pending minor revisions.

Abstract: the abstract appropriately summarize the manuscript without discrepancies between the abstract and the remainder of the manuscript.

Keywords: inadequate. Please, reach 5 keywords.

Reference: please, follow my suggestions.

Paper

On some aspects, the authors should address:

1)You defined SBLE as “a hypoechoic, well-defined band, located immediately under the epidermis layer and corresponding to the papillary dermis”.

I advise you to specify the difference between papillary and reticulate dermis in terms of collagen composition and vascularization. You can find the latter aspect in the following article, which you must cite, in which you can also find an updated description of the ultrasound and ecovascular aspects of normal and pathological dermal vascularization evaluated with microcirculation software.

-Seeing the unseen with superb microvascular imaging: Ultrasound depiction of normal dermis vessels. J Clin Ultrasound. 2022 Jan;50(1):121-127. doi: 10.1002/jcu.23068. Epub 2021 Nov 10. PMID: 34761407.

2)It would be interesting if you included a schematic diagram depicting normal skin and skin with SLEB assessed by ultrasound (also Doppler US).

3)I did not find a reference in the text to Table 1.

4)Regardless of the systematic review, ultrasound images should be included. Do you use high frequency probes in your daily activity? Do you have probes with frequencies higher than 24 MHz? Please provide for it.

Figures:

-I have not found any images. I would recommend you to insert high resolution ultrasound images of normal and pathological cases.

-Why don't you try inserting images at different frequencies? For example, 18 MHz, 24 MHz and very-high US images (>24 if you have).

Author Response

Thank you for your comments and suggestions. They are really very useful and added value to our manuscript. Unfortunately, we only have the ultrasound with the attached frequency in the work, as a result we could not add more types of frequencies at the moment. We are in the process of procuring an additional ultrasound machine with a different frequency, so we will take this suggestion into account for future articles. Regarding the suggestions related to the papillary dermis component, I have introduced additional explanations. Also, there really weren't enough keywords, so I added a few more suitable to the theme. Thank you again for the suggestions!

Reviewer 2 Report

The authors made a systematic review on the subepidermal low echogenic band using high-frequency ultrasonography (LF-USG) in the dermatological practice. High reproducibility, non-invasiveness and short diagnostic time are its advantages. Subepidermal low echogenic band (SLEB) as a relatively newly described  parameter can be utilized to quantify intrinsic and extrinsic skin aging as well as inflammatory processes. SLEBs value in the diagnosis and monitoring of the treatment of some inflammatory and non-inflammatory dermatological conditions has been reviewed.

In abstract, high-frequency ultrasound should be replaced by high-frequency ultrasonography to present LF-USG.

Representative photos should be included to illustrate the application values of SLEB in different typical cases.

Requirement or technical development for better clinical applications may also be added.

The authors should also edit the manuscript, especially the comma at the end of each sentence and space between the sentences.

Author Response

Thank you for your valuables comments. We have made all the changes according with your suggestions. I have also introduced some skin ultrasonography images performed with a 20MHz transducer with normal skin, with psoriasis and atopic dermatitis in which the different structures can be highlighted, including SLEB.

Reviewer 3 Report

none

Author Response

Thank you for your comments and evaluation. 

Round 2

Reviewer 2 Report

revision is good for publication